# Functional independence of endogenous μ- and δ-opioid receptors co-expressed in cholinergic interneurons

**Seksiri Arttamangkul[1]\*, Emily J Platt[2], James Carroll[3], David Farrens[2]**

[1]The Vollum Institute, Oregon Health & Science University, Portland, United States; [2]Department of Chemical Physiology and Biochemistry, School of Medicine, Oregon Health & Science University, Portland, United States; [3]Program in Biomedical Sciences, Department of Chemical Physiology and Biochemistry, Oregon Health & Science University, Portland, United States

**\*For correspondence:**
arttaman@ohsu.edu

**Competing interest:** The authors declare that no competing interests exist.

**Abstract** Class A G-protein-coupled receptors (GPCRs) normally function as monomers, although evidence from heterologous expression systems suggests that they may sometimes form homodimers and/or heterodimers. This study aims to evaluate possible functional interplay of endogenous μ- and δ-opioid receptors (MORs and DORs) in mouse neurons. Detecting GPCR dimers in native tissues, however, has been challenging. Previously, MORs and DORs co-expressed in transfected cells have been reported to form heterodimers, and their possible co-localization in neurons has been studied in knock-in mice expressing genetically engineered receptors fused to fluorescent proteins. Here, we find that single cholinergic neurons in the mouse striatum endogenously express both MORs and DORs. The receptors on neurons from live brain slices were fluorescently labeled with a ligand-directed labeling reagent, NAI-A594. The selective activation of MORs and DORs, with DAMGO (μ-agonist) and deltorphin (δ-agonist) inhibited spontaneous firing in all cells examined. In the continued presence of agonist, the firing rate returned to baseline as the result of receptor desensitization with the application of deltorphin but was less observed with the application of DAMGO. In addition, agonist-induced internalization of DORs but not MORs was detected. When MORs and DORs were activated simultaneously with [Met$^5$]-enkephalin, desensitization of MORs was facilitated but internalization was not increased. Together, these results indicate that while MORs and DORs are expressed in single striatal cholinergic interneurons, the two receptors function independently.

## Introduction

Opioid receptors are a family of GPCRs targeted by endogenous peptides and exogenous opiate drugs. Three opioid receptor-subtypes, μ (MOR), δ (DOR), and $\kappa$ (KOR) are known to be distributed in many brain areas (*Le Merrer et al., 2009*). The co-expression of opioid receptor-subtypes in individual neurons has steadily gained attention because possible cross-interactions between the receptors could provide new pharmacological targets for analgesia while avoiding the side effects associated with the activation of a single opioid receptor subtype (*Fujita et al., 2015*). Although there is evidence from biochemical and fluorescent imaging studies to support the existence of MOR-DOR heterodimers (*Cahill and Ong, 2018*), those studies required the use of heterologous expression and genetically modified receptors to enable experimental detection. In addition, an increasing number of studies suggest that only a small population of dimers exist and interact in a time scale of milliseconds, resulting in a proposed 'kiss-and-run' model of receptor dimer interaction (*Gurevich and Gurevich, 2008*; *Kasai and Kusumi, 2014*; *Calebiro and Sungaworn, 2018*). Thus, there is still no

clear evidence that endogenously expressed MORs and DORs form stable heterodimers. The major challenge for testing these possible interactions in native tissue is the fact that opioid receptors are generally expressed in low densities in neuronal tissues, thus detection of subcellular co-localization is often ambiguous. Functional readout using electrophysiological measurement is a powerful tool to detect MOR-DOR co-expression in single neurons but there are few such studies (*Egan and North, 1981*; *Chieng et al., 2006*).

The striatum is a subcortical structure in the forebrain that plays an important role in motivation, sensorimotor function, goal-directed learning, and drug addiction (*Kreitzer, 2009*; *Castro and Bruchas, 2019*). The largest portion of neurons are medium spiny GABAergic projecting neurons that are also known to produce opioid peptides including enkephalin (μ-and δ-agonists) and dynorphin ($\kappa$ -agonists). The striatum consists of a heterogeneous structure where all opioid receptor-subtypes have been located by various techniques (*Le Merrer et al., 2009*). MOR-binding sites are primarily detected in the striosomal patches, whereas DOR and KOR binding sites are found in both matrix and patches (*Herkenham and Pert, 1981*; *Tempel and Zukin, 1987*; *Kitchen et al., 1997*; *Banghart et al., 2015*). Cholinergic interneurons (ChIs) comprise only a small population (1–2% of all neurons) and are scattered throughout the striatum. These neurons function as an important regulator of the synaptic activity within the striatum including the local release of dopamine (*Cai et al., 2021*; *Yorgason et al., 2017*; *Threlfell et al., 2012*; *Cachope et al., 2012*). ChIs are spontaneously active neurons that release acetylcholine in brain slices and in vivo (*Wilson et al., 1990*; *Bennett and Wilson, 1999*; *Zhou et al., 2002*). The firing rate of these neurons is inhibited, and the membrane potential is hyperpolarized following the application of δ- but not μ-agonists (*Jiang and North, 1992*). Other studies, however, report the presence of functional MORs in these neurons (*Jabourian et al., 2005*; *Britt and McGehee, 2008*; *Ponterio et al., 2013*; *Mamaligas et al., 2016*; *Muñoz et al., 2018*). It is possible that various ChI subpopulations contain segregated MORs and DORs (*Jabourian et al., 2005*; *Britt and McGehee, 2008*; *Laurent et al., 2012*). To date, there is no evidence demonstrating that MORs and DORs are co-localized on ChIs.

Thus, in the present study, we used a novel labeling approach combined with image analysis and electrophysiological techniques to investigate the distribution of MORs and DORs on ChI neurons in the striatum. To enable live-cell imaging, we used a ligand-directed labeling agent, NAI-A594, to specifically attach a fluorophore to both MORs and DORs on the plasma membrane of single ChIs in brain slice preparations. The selective antagonists at DOR (SDM25N) or MOR (CTAP) were used to differentiate the two receptors and assess the degree of MOR and DOR expression. The inhibition of spontaneous firing using the selective μ-agonist (DAMGO) and δ-agonist (deltorphin II) confirmed that both MORs and DORs were present and functional on single neurons. [Met$^5$]-enkephalin (ME) and morphine also inhibited the firing activity of ChIs. As expected, ME interacted with both receptors while morphine selectively activated MORs. Interestingly, while the extracellular recordings showed activation at each receptor fully inhibited firing activity of the neuron, in the continued presence of agonists substantial desensitization of DORs, but not MORs was observed. Similarly, only the DORs showed agonist-induced receptor internalization. Co-activation of the receptors by exogenous application of ME increased MOR desensitization. Collectively, the results in this study indicate that co-localized MOR-DOR function autonomously and there is little evidence for stable heterodimer formation, at least at functional level. Nonetheless, our data do not rule out possible downstream functional interactions that may occur when both receptors are simultaneously activated.

## Results
### Ligand-directed labeling of opioid receptors

Naltrexamine-acylimidazole-alexa594 (NAI-A594; *Figure 1A–a*) is a labeling reagent that has been shown to specifically bind opioid receptors and then covalently tag them with the fluorophore Alexa-594 to MORs in brain slices (*Arttamangkul et al., 2019*). The labeling of this molecule is based on traceless affinity labeling approach (*Hayashi and Hamachi, 2012*; *Shiraiwa et al., 2020*), in which the naltrexamine moiety acts as a ligand that guides specific binding to opioid receptors. Once bound, the fluorophore is transferred to the receptor by the reaction of the acylimidazole group with an amino acid side chain nucleophile, and at the same time, the guide ligand (naltrexamine) is cleaved and released from the binding pocket (*Figure 1A–b*).

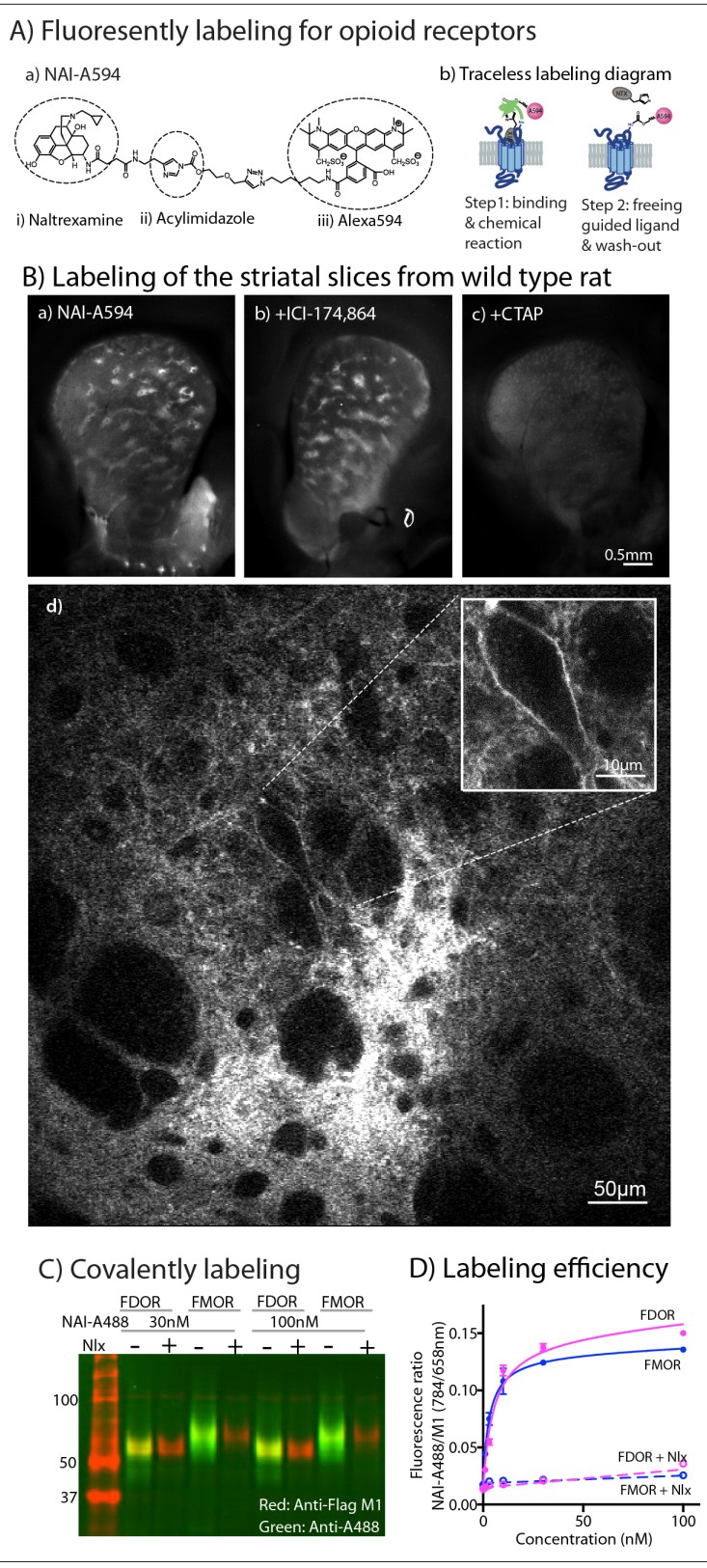

**Figure 1.** Labeling of naltrexamine-acylimidazole derivatives. (**A**) Covalent labeling of opioid receptors with NAI compounds. (**a**) Chemical structure of naltrexamine-acylimidazole-alexa594 (NAI-A594). (**b**) Diagram of traceless labeling shown in two steps. Step 1, NAI compound binds and reacts with the receptor. Step 2, washing of naltrexamine moiety released from the reaction. (**B**) Live images of rat striatum incubated for 1 hr in (**a**) 100 nM

*Figure 1 continued on next page*

*Figure 1 continued*

NAI-A594, (**b**) 100 nM NAI-A594 plus 1 µM ICI-174,864, (**c**) 100 nM NAI-A594 plus 1 µM CTAP, and (**d**) zoom area of patch-like structure in the dorsal part of striatum showing a large neuron believed to be the cholinergic interneuron at the boundary of the patch. (**C**) Near infrared western blots of HEK293 cells expressing FDOR and FMOR labeled with 30 and 100 nM NAI-A488 in the absence and presence of 10 µM naloxone, shown in a merge channel. The FLAG-epitope on opioid receptors was detected with anti-FLAG M1 (red), and the NAI-A488 modification of the receptors was identified with rabbit anti-Alexa Fluor 488 antibody (green). (**D**) Concentration-dependent curves of FDOR and FMOR labeled with NAI-A488. The fluorescence intensity ratios of NAI-A488 (784 nm of secondary antibody to IR dye CW800) over anti-Flag M1 (658 nm of secondary antibody conjugated to Alexa680) were plotted against NAI-A488 concentrations. The graph shows labeling curves performed in triplicate from one of three experiments. Data are shown as mean ± SEM.

The online version of this article includes the following figure supplement(s) for figure 1:

**Source data 1.** Uncropped merged gel of western analysis of FDOR and FMOR labeled with NAI-A488.

**Source data 2.** Quantification of labeling efficiency by on-cell western analysis.

**Figure supplement 1.** Covalently labeling and quantification of FDOR and FMOR in HEK293 cells.

**Figure supplement 1—source data 1.** Uncropped red channel (anti-Flag M1) gel of western analysis of FDOR and FMOR labeled with NAI-A488.

**Figure supplement 1—source data 2.** Uncropped green channel (rabbit anti-A488) gel of western analysis of FDOR and FMOR labeled with NAI-A488.

**Figure supplement 1—source data 3.** The raw image of on-cell western analysis of labeled receptors (anti-Flag M1).

**Figure supplement 1—source data 4.** The raw image of on-cell western analysis of labeled receptors (rabbit anti-A488).

**Figure supplement 1—source data 5.** The raw image of on-cell western analysis of labeled receptors (merge).

The labeling of striatal slices from wild-type rats with NAI-A594 revealed striosomal patches, regions that are known to contain abundant MORs (*Figure 1B–a* and *Arttamangkul et al., 2019*). The MOR-binding sites in the patches persisted when DORs were blocked with a selective antagonist ICI-174,864 (*Figure 1B-b*). In contrast, fluorescent labeling of the whole striatum was greatly diminished when the MOR selective antagonist, CTAP, was included in NAI-A594 solution (*Figure 1B–c*). These results suggest that most of the labeling in the striatum were at MORs. Large neurons with a few dendrites near the boundary of the striosomal patches were also fluorescently tagged (*Figure 1B–d*). The cell morphology and location of these neurons suggested that they were most likely ChI neurons (*Brimblecombe and Cragg, 2016*). The labeled receptors on these large neurons, however, could be either MORs or DORs, due to the ability of NAI-A594 to label both receptors (*Arttamangkul et al., 2019*).

To verify the comparable labeling of MORs and DORs with NAI-A594, a series of biochemical experiments were done using Flag-tagged MOR (FMOR) and Flag-tagged DOR (FDOR) cells. Naltrexamine-acylimidazole-alexa488 (NAI-A488), was used to detect the labeled receptors with an anti-Alexa488 antibody, and an anti-Flag antibody was used as a control for comparison in a western blot analysis. The results showed that NAI-A488 at 30 and 100 nM reacted similarly with both receptors (*Figure 1C*, *Figure 1—figure supplement 1A* and *Figure 1—figure supplement 1—source data 1* and *Figure 1—figure supplement 1—source data 2*, for red and green bands, respectively). The labeling of FDOR and FMOR by NAI-A488 was specific, as it could be completely blocked by addition of the opioid antagonist naloxone (*Figure 1C*). The fact that Alexa488-labeled receptors are detected even after denaturing gel electrophoresis confirms that the fluorophore is covalently bound to the receptors. The ability of NAI-A488 to label FDOR and FMOR was further analyzed by comparing concentration labeling curves of each receptor using an on-cell western assay (*Figure 1D*, *Figure 1—figure supplement 1—source data 3*, *Figure 1—figure supplement 1—source data 4*, *Figure 1—figure supplement 1—source data 5*). The labeling curves for NAI-A488 reacted FDORs and FMORs were similar and saturated at about 100 nM. The labeling coefficients were also comparable, with apparent Kd values of 7.1 ± 2.8 nM for FDOR and 3.3 ± 0.7 nM for FMOR (two-tailed, unpaired t-test, p = 0.2543, n = 3 experiments performed in triplicate). Thus, the fluorescent NAI compound effectively and covalently labeled MORs and DORs.

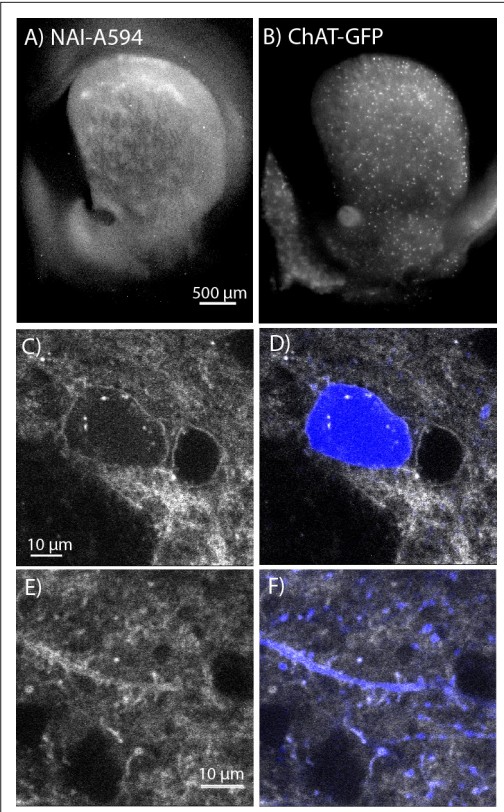

**Figure 2.** In situ NAI-A594 labeling of ChI neurons in the striatum of ChAT-GFP mice. (**A**) An area where a fluorescent patch is clearly observed in the striatum of ChAT-GFP after the staining with NAI-A594. (**B**) Distribution of GFP-positive neurons in the same section of brain slice in (**A**). (**C–F**) The labeling of NAI-A594 is clearly observed along plasma membrane staining of cell body, dendrites, and terminals. The gray outline of two cells in (**C**) and the merged channel shows GFP (blue) in (**D**) indicate that the other neuron, presumably MSN is also stained with NAI-A594. Neurites containing GFP are visible after staining (**E**). Fluorescence of neurites is observed throughout the area. The merged image (**F**) shows NAI-A594 labeled both positive and negative GFP neurites.

## Labeling of MORs and DORs on cholinergic interneurons with NAI-A594

To determine which opioid receptors are present on the striatal ChIs, brain slices from transgenic ChAT(BAC)-eGFP mice were used to identify ChIs. As previously observed, patch-like structures were present in the striatum of the ChAT-GFP mice following incubation with NAI-A594 (*Figure 2A*). The ChAT-GFP neurons were found scattered throughout the striatum (*Figure 2B*). The plasma membrane of GFP-positive perikarya, dendrites and axon terminals were also Alexa 594 positive (*Figure 2C&E* for NAI-A594 staining and D&F for the merged images of NAI-A594 labeling and GFP). There are GFP-negative neurons, presumably medium spiny neurons that were also labeled with NAI-A594 (*Figure 2C&D*).

Live-cell imaging experiments revealed all GFP-positive neurons reacted with NAI-A594 (n = 246 from seven male and six female mice). *Figure 3A* shows example images of GFP-positive cells labeled with NAI-A594 alone, and NAI-A594 in the presence of µ-selective antagonist CTAP (1 µM) such that DORs alone were identified. In experiments with NAI-A594 in the presence of δ-selective antagonist SDM25N (0.5 µM), MORs were specifically detected. The results show a partial decrease of the fluorescence on the plasma membrane of CTAP-treated neurons as compared to experiments with NAI-A594 alone (see *Figure 3A–b* and *Figure 3A-a*). In the presence of SDM25N, the fluorescence staining was further reduced as compared to CTAP-treated cells (*Figure 3A–c* and *Figure 3A-b*). Finally, the combination of SDM25N and CTAP completely blocked the labeling of ChAT-GFP neurons (*Figure 3A–d*).

To estimate the relative amount of MORs and DORs on ChAT-GFP neurons, the fluorescent intensity of Alexa594 on an individual ChAT-GFP neuron was measured in different parts of the striatum. Results are presented as mean fluorescence of Alexa 594 along the plasma membrane of the cell body (pink line derived from rendering of the cytoplasmic GFP signal in *Figure 3B*). The results from male and female mice were not different (*Figure 3—figure supplement 1A*: *Figure 3—figure supplement 1—source data 1*), thus the data were pooled. Average mean fluorescence intensity of NAI-A594-labeled cells in the dorsal striatum was ~20 % higher than in ventral striatum (mean fluorescence± SD = 170.7 ± 57.6 for dorsal, n = 52 and 140.9 ± 47.6 for ventral n = 47; ordinary one-way ANOVA, Sidak's test, p = 0.0100 and n = numbers of GFP-positive neurons, *Figure 3C*, *Figure 3—source data 1*). When DOR was measured by co-incubating with NAI-A594 plus CTAP, the fluorescent intensity on neurons in the dorsal striatum was reduced to 68 % of that observed with NAI-A594 alone (mean fluorescence± SD = 115.5 ± 49.7, n = 41 as compared to the mean fluorescence of NAI-A594 = 170.7 ± 57.6 for dorsal, n = 52; p < 0.0001, *Figure 3C*). Similarly, the fluorescence of neurons in the ventral striatum was reduced to 61 % of NAI-A594 alone (mean fluorescence± SD = 85.8 ± 47.3, n = 35 as compared to the mean fluorescence of NAI-A594 = 140.9 ± 47.6 for ventral, n = 47; p < 0.0001,

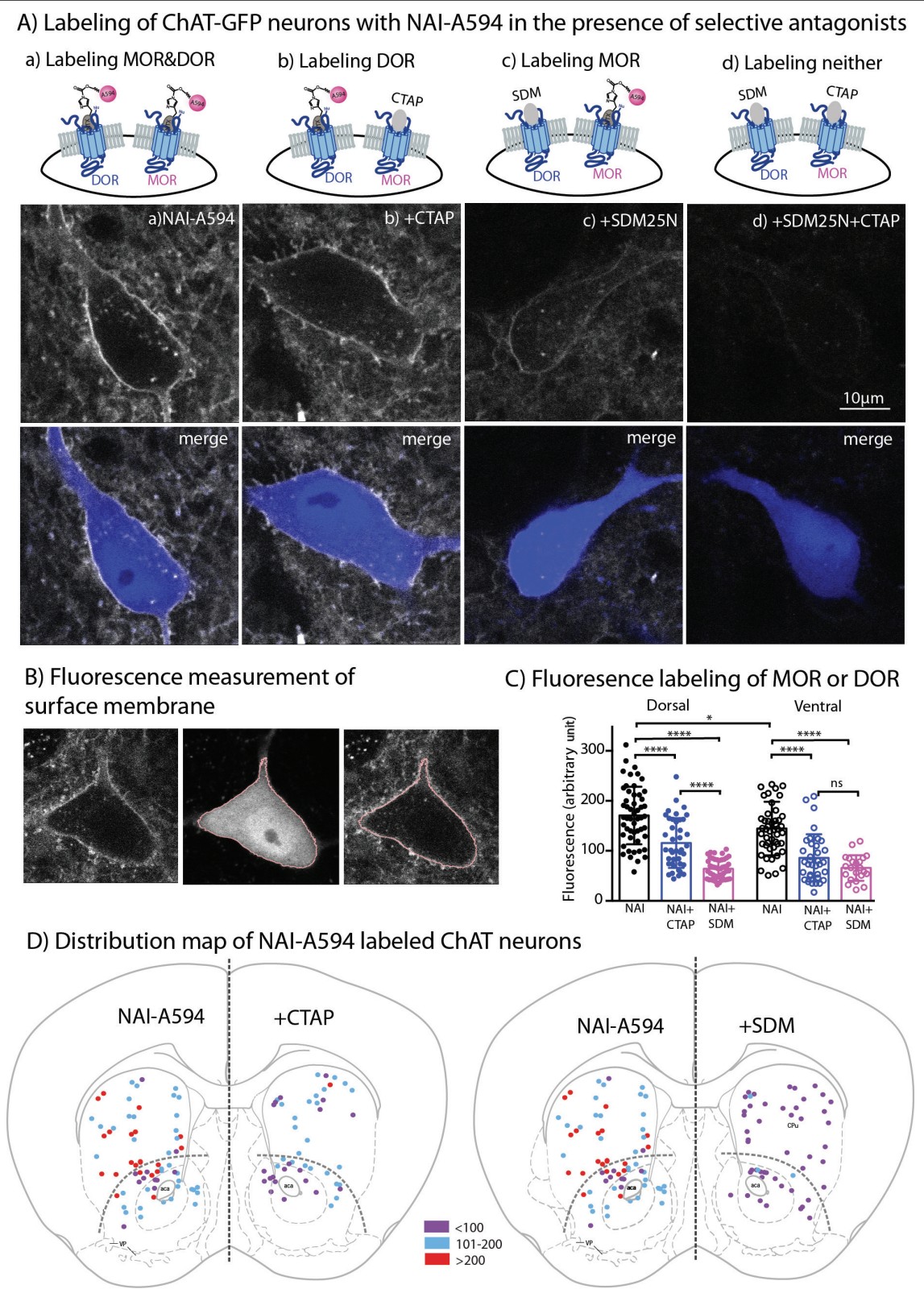

**Figure 3.** Fluorescent intensities of NAI-A594 labeling of endogenous MOR or DOR receptors. (**A**) Examples of single ChAT-GFP positive cell showing staining of NAI-A594 under different conditions: (**a**) NAI-A594, (**b**), NAI-A594+ CTAP, (**c**) NAI-A594+ SDM25 N, and (**d**) NAI-A594+ CTAP and SDM25N. The upper panel are images from Alexa594 channel, and the bottom panel are the merged images of GFP (blue) and Alexa594 (gray). (**B**) Diagram images illustrate the method of measurement of surface membrane fluorescent intensity. Left is the raw signal of a neuron labeled with NAI-A594.

*Figure 3 continued on next page*

*Figure 3 continued*

Middle shows outline boundary derived from GFP signal and thresholding using ImageJ (details in Materials and methods). Right, the fluorescent intensities of Alexa594 channel are measured along the outline drawn in the GFP channel and superimposed onto the raw image. The lines drawn here are enlarged for illustration. The true thickness of line is ~0.08 μm or one pixel. (**C**) Summarized results of fluorescent intensities in each labeling condition. Dorsal and ventral areas are defined according to the mouse brain atlas (*Franklin and Paxinos, 2007*). All data are shown in mean ± SD, and statistical analyses using ordinary one-way ANOVA and Sidak's post-test for multiple comparisons. (**D**) Distribution maps of the observed ChIs, in which fluorescent intensities were measured in different conditions. Data are presented as combined results from both male and female. Color codes represent the range of fluorescent intensities (in Arbitrary Units, AU) shown in (**C**).

The online version of this article includes the following figure supplement(s) for figure 3:

**Source data 1.** Membrane fluorescent intensities of A594-labeled both MOR and DOR (NAI), DOR (NAI+ CTAP), and MOR (NAI+ SDM25 N).

**Figure supplement 1.** Fluorescent labeling of MOR or DOR.

**Figure supplement 1—source data 1.** Membrane fluorescent intensities of A594-labeled MOR and DOR (NAI), DOR (NAI+ CTAP) or MOR (NAI+ SDM25 N) from male and female ChAT-GFP mice.

**Figure supplement 1—source data 2.** Membrane fluorescent intensities of A594-labeled neurons from MOR-KO and ChAT-GFP mice.

*Figure 3C*). Fluorescence of Alexa594-labeled ChAT-GFP neurons in the presence of CTAP was found to be similar to the labeling of the cholinergic neurons from MOR knockout mice (*Figure 3— source data 1*: *Figure 3—figure supplement 1—source data 2*), suggesting that the blockade of NAI-A594 labeling at MORs by CTAP was sufficient. Interestingly, when the detection of MOR was determined by NAI-A594 in the presence of SDM25N, the fluorescent intensity of labeled cells was only ~37 % and ~ 47 % of that observed with NAI-A594 alone in the dorsal and ventral striatum, respectively (mean fluorescence± SD = 63.9 ± 19.5, n = 48 as compared to the mean fluorescence of NAI-A594 = 170.7 ± 57.6, n = 52 for dorsal, and 65.7 ± 25.5, n = 22 as compared to the mean fluorescence of NAI-A594 = 140.9 ± 47.6, n = 47 for ventral; p < 0.0001 for both, *Figure 3C*). There was a large difference between fluorescent staining of DOR and MOR in dorsal part of the striatum (115.5 ± 49.7, n = 41 for DOR *vs*. 63.9 ± 19.5, n = 48 for MOR, one-way ANOVA, p < 0.0001), while the relative fluorescence of DOR and MOR in the ventral striatum was comparable (85.8 ± 47.3, n = 35 for DOR *vs*. 65.7 ± 25.5, n = 22 for MOR, p = 0.612, *Figure 3C*). Thus, these results indicate that: (i) cholinergic interneurons contain both MORs and DORs and (ii) expression of DORs appears to be higher than MORs in the dorsal striatum but not different from MORs in the ventral striatum. *Figure 3D* shows the distribution map of NAI-A594 labeled ChAT-GFP-positive neurons.

## Inhibition of ChI spontaneous firing by MOR and DOR agonists

Cholinergic interneurons are spontaneously active in mouse brain slice preparations with a firing rate of 1–2 Hz (*Ponterio et al., 2013*). Cell attached extracellular recordings were used to measure the firing activity and the action of MOR and DOR agonists. All recordings were made in the ventral striatum because of the similar expression of MOR and DOR. The average spontaneous firing rate of the GFP-positive neurons was 1.5 ± 0.1 Hz (n = 9 cells from three male and three female mice). *Figure 4* shows a representative trace and time-course of firing rate (15- second binning) during

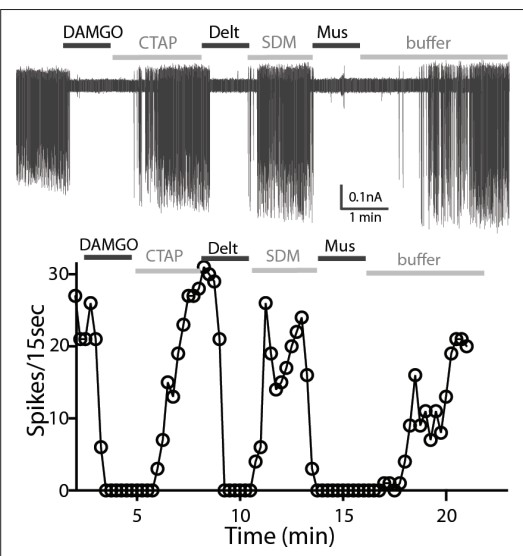

**Figure 4.** A representative trace of spontaneous firing shows regular spiking at baseline, and a complete stop of firing (100 % inhibition) occurs after application of DAMGO (1 μM, 1 min). The firing returns to baseline with application of CTAP (1 μM). The subsequent application of deltorphin (1 μM, 1 min) also causes 100 % inhibition and this effect is reversed with SDM25N (0.5 μM). Lastly, muscarine (10 μM) causes 100 % inhibition of firing. The graph below the trace shows the time course of the experiment plotted against the number of spikes at 15 s binning.

**Table 1.** Activation and desensitization by opioids.

| Agonists | Inhibition of firing (%)* | Desensitization (%) |
|---|---|---|
| DAMGO (10 µM) | 98.7 ± 0.9 (n = 11, 4 M,3F) | 32.1 ± 5.8 (n = 11)† |
| DAMGO (1 µM) | 98.0 ± 1.2 (n = 7, 2 M,4F) | |
| DAMGO (1 µM) after Deltorphin (1 µM)‡ | 93.5 ± 6.1 (n = 5, 1 M,2F) | |
| DAMGO (1 µM) after ME (1 or 10 µM)‡ | 36.4 ± 6.0 (n = 7, 3 M,3F)§ | |
| Deltorphin (1 µM) | 96.6 ± 2.7 (n = 10, 2 M,2F) | 86.7 ± 6.4 (n = 7) |
| Deltorphin (1 µM) after DAMGO (10 µM)‡ | 89.8 ± 5.8 (n = 6, 3 M,2F) | |
| Deltorphin (1 µM) after ME (10 µM)‡ | 18.3 ± 3.4(n = 5, 1 M,2F)¶ | |
| ME (10 µM) | 99.4 ± 0.6 (n = 11, 3 M,3F) | 83.0 ± 6.0 (n = 8) |
| ME (1 µM) | 92.6 ± 2.6 (n = 12, 6 M,2F) | 79.9 ± 9.9 (n = 5) |
| ME (1 µM)+ CTAP (1 µM) | 97.7 ± 1.4 (n = 6, 2 M,1F) | 76.8 ± 8.4 (n = 6) |
| ME (1 µM)+ SDM (1 µM) | 97.3 ± 1.7 (n = 6, 4 M) | 26.1 ± 8.8 (n = 5) |
| Morphine (10 µM) | 79.4 ± 5.0 (n = 10, 4 M) | |
| Morphine (1 µM) | 48.9 ± 8.3 (n = 8, 4 M) | |
| Morphine (10 µM)+ CTAP (1 µM) | 10.0 ± 3.7 (n = 7, 4 M) | |

*n = number of cells (one cell per slice), and numbers of male (M) and female (F) used in the experiments are shown in parentheses.

†p < 0.0001 compared to desensitization induced by Deltorphin (1 µM).

‡Inhibition of firing after agonist-induced desensitization.

§p < 0.0001 compared to firing inhibition of DAMGO (1 µM).

¶p < 0.0001 compared to firing inhibition of Deltorphin (1 µM).

The online version of this article includes the following source code for table 1:

**Source data 1.** Spontaneous firing inhibition (%) and desensitization induced by agonists.

agonist and antagonist application. The µ-agonist, DAMGO (1 µM) completely inhibited firing, and this inhibition was reversed following application of CTAP (1 µM). Subsequent application of the δ-agonist, deltorphin (1 µM) also stopped firing and this was reversed by SDM25N (0.5 µM, *Figure 4A*). The inhibition of firing induced by muscarine (10 µM) was included as a control. The results indicate that ChIs co-express functional MORs and DORs.

The inhibition of firing induced by opioid agonists was also studied in neurons from wild-type C57/B6 mice. ChIs were identified by the cell morphology and regular firing activity. As in experiments with the slices from the ChAT-GFP mice, neurons in slices from wild-type animals were responsive to consecutive application of DAMGO and deltorphin (n = 19 cells, from four female and six male mice, data not shown). Each opioid agonist decreased the spontaneous firing and is reported as a percent inhibition, calculated by measuring the firing activity before (baseline) and at 1–2 min following agonist application (*Table 1*).

The firing frequency of ChIs from wild-type mice was similar to that of ChAT-GFP (1.4 ± 0.1 Hz, n = 67 cells from fourteen male and eleven female mice, compared to the firing rate of ChAT-GFP neurons of 1.5 ± 0.1 Hz, n = 9 cells, two-tailed, unpaired t-test, p = 0.6490). DAMGO (1 µM) caused an inhibition of firing activity by 98.0% ± 1.2% (n = 7) and increasing the concentration to (10 µM) had no further effect (*Table 1*). Deltorphin (1 µM) also caused a total inhibition of firing (96.6% ± 2.7% n = 10). Application of a saturating concentration of [Met⁵]-enkephalin (ME, 10 µM) resulted in a complete inhibition of the spontaneous firing (99.4% ± 0.6%, n = 11) and ME (1 µM) reduced the firing by 92.6% ± 2.6% of baseline (n = 12). Because ME acts on MORs and DORs (*Gomes et al., 2020*), selective antagonists were used to identify the receptor. When CTAP was used to block activity of MORs, ME (1 µM) inhibited the firing activity by 97.7% ± 1.4% (n = 6) (*Table 1*, *Table 1—source data 1*). When DORs were blocked with SDM25N, ME (1 µM) reduced the firing rate by 97.3% ± 1.7% (n = 6, *Table 1*). Thus, as expected, ME acted on both receptors. Morphine (10 µM) only partially decreased the firing by 79.4% ± 5.0% (n = 10) and morphine (1 µM) reduced the firing rate by 48.9% ± 8.3% (n

= 8) (*Table 1*). In addition, CTAP blocked the inhibition induced by morphine (10.0% ± 3.7% inhibition in 10 µM morphine +1 µM CTAP, n = 7) (*Table 1*). The results from these experiments provide further evidence of individual activation of MORs and DORs in ChIs by various opioids.

## Receptor desensitization

The application of saturating concentrations of efficacious agonist commonly results in the desensitization of MORs and DORs in many brain areas (*Williams et al., 2013*; *Gendron et al., 2016*). We next used the inhibition of firing rate induced by DAMGO and deltorphin to determine the extent of MOR and DOR desensitization. Desensitization was calculated from the difference of percent inhibition of firing inhibition at 5 min and the percent maximal inhibition at 1–2 min. Deltorphin (1 µM) initially stopped the firing and within 3–5 min the firing returned toward baseline (*Figure 5A*, top trace). Thus, deltorphin triggered significant desensitization to 86.7% ± 6.4% (n = 7, *Table 1*). In contrast, DAMGO (10 µM) induced only 32.1% ± 5.8 % desensitization (n = 11, *Figure 5A* bottom trace, *Table 1*). Together, these results indicate that DORs significantly desensitize within 5 min whereas MORs are more resistant to desensitization (*Figure 5B*, *Figure 5—source data 1*, one-way ANOVA with Sidak's multiple comparisons test, p < 0.0001).

The interaction between the desensitization of DORs and activation of MORs was next examined by the application of deltorphin (1 µM, 5 min) followed by DAMGO (1 µM, *Figure 5C* top trace). Desensitization caused by deltorphin (1 µM) did not change the ability of DAMGO (1 µM) to fully inhibit the firing rate (93.5% ± 6.1% inhibition after deltorphin, n = 5 compared to 98.0% ± 1.2% inhibition without deltorphin pre-treatment, n = 7, one-way ANOVA, Sidak's multiple comparions, p = 0.9343, *Figure 5F*, *Figure 5—source data 3*). The reverse experiment where DAMGO (10 µM) was applied first followed by deltorphin (1 µM) was examined next (*Figure 5C* -middle trace and *Figure 5F*). There was a small return of firing due to DAMGO-induced desensitization and the subsequent application of deltorphin caused an inhibition. The inhibition induced by deltorphin after DAMGO was not statistically different from the inhibition of deltorphin without DAMGO pre-treatment (81.6% ± 6.2% of DAMGO pre-desensitized, n = 5, as compared to 96.6% ± 2.7% of control, n = 10, p = 0.6687, *Figure 5F*). These results suggest that MORs and DORs are functionally independent and that there is no major long-lasting cross desensitization between the receptors by selective agonists.

Next, the degree of desensitization at MORs and DORs was examined with the simultaneous activation of both receptors. ME was used in the experiments because it has similar binding affinity at both receptors (*Gomes et al., 2020*), and it completely inhibited firing of ChIs (*Table 1*). Similar to the action of deltorphin, ME (1 µM) plus CTAP fully inhibited the firing that returned to baseline during a 5 -min application resulting in 76.8% ± 8.4% desensitization of DOR (n = 6, *Figure 5D* top trace, *Figure 5B*, p = 0.9218, *Table 1*). The desensitization of MORs with ME (1 µM) plus SDM25N was much less than ME plus CTAP (26.1% ± 8.8%, n = 5 for ME plus SDM25N *vs.* 76.8% ± 8.4% for ME plus CTAP *Figure 5D* middle trace and *Figure 5B*, one-way ANOVA, p = 0.0040). Additionally, time course analyses show MOR desensitization caused by ME plus SDM25N was slower than DOR desensitization caused by ME plus CTAP (two-way ANOVA, p < 0.0001, *Figure 5E*, *Figure 5—source data 2*). Thus, the results indicate that ME will selectively activate and desensitize MORs or DORs if the other receptor is blocked by a selective antagonist. In experiments where no antagonists were used, ME (1 µM) caused 79.9% ± 9.9% desensitization (n = 5, *Figure 5D* bottom trace and *Figure 5B*) that was similar to ME (10 µM, 83.0% ± 6.0%, n = 8, p = 0.9999). The degree of desensitization caused by ME (1 µM) was not different from that caused by ME plus CTAP or deltorphin (*Figure 5B*, one-way ANOVA, Sidak's multiple comparisons, p = 0.9999 for ME (1 µM) *vs.* ME plus CTAP, or p = 0.9913 for ME (1 µM) *vs.* deltorphin).

These results were surprising because the above findings predict that activation of MORs by ME should sustain inhibition of firing activity longer, since MOR only weakly desensitizes when selectively activated. The time-course analysis of desensitization during 5 -min application of ME in different conditions reveals that desensitization caused by ME activation at both receptors started slower than the desensitization of DOR caused by ME plus CTAP (*Figure 5E*, two-way ANOVA, p = 0.0109 for different treatment and p = 0.0001 for time of agonist application, as compared to ME 1 µM). Conversely, the desensitization by ME alone was faster and more complete than desensitization of MOR caused by ME plus SDM25N (*Figure 5E*, two-way ANOVA, p = 0.0001 for different treatment and p = 0.0040 for time of agonist application, as compared to ME 1 µM).

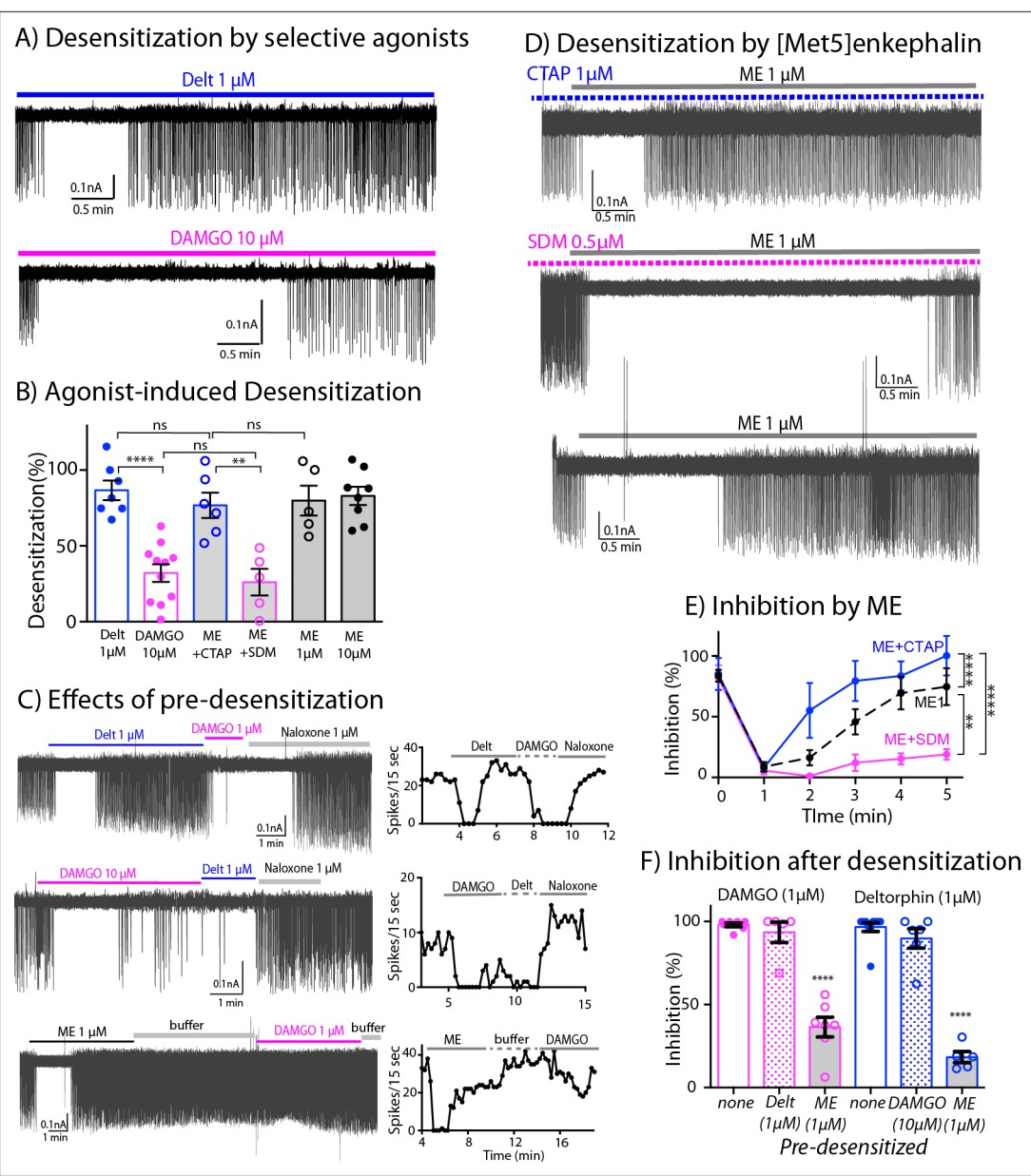

**Figure 5.** Differential desensitization by selective activation of MOR and DOR. (**A**) *Top* trace shows spontaneous firing of ChI neurons during 5 min application of the DOR selective agonist Deltorphin (1 μM) that caused 100 % inhibition of the firing during the first 1–2 min and the firing activity gradually returned to nearly baseline during the last 3 min of continuous application of the agonist, an indication of desensitization. *Bottom* trace. In contrast, the selective agonist DAMGO caused the persistent inhibition of the firing throughout its 5 min application. (**B**) Summary results of desensitization by Deltorphin, DAMGO, ME (1 and 10 μM) and ME (1 μM) with selective antagonists, CTAP (1 μM) and SDM25N (0.5 μM). (**C**) *Top* trace, desensitization of DORs by deltorphin did not change the ability of DAMGO to stop the firing that was reversed by naloxone. *Middle* trace shows similar experiment in which DAMGO caused little desensitization and subsequent application of deltorphin also inhibited the firing. *Bottom* trace illustrates the experiment when desensitization by ME decreased the ability of DAMGO to inhibit the firing. Time courses of each experiment plotted as the number of spikes binned every 15 s are shown next to the activity traces. (**D**) Desensitization by ME is very different when co-incubated with CTAP or SDM25N. *Top* trace shows ME desensitization in CTAP developed quickly. *Middle* trace shows no or little desensitization occurred during application of ME plus SDM25N. *Bottom* trace represents desensitization caused by ME alone. (**E**) The graph shows change of % inhibition of firing activity during 5 -min application of ME in different conditions, Two-way ANOVA showed ** p = 0.0040 and **** p < 0.0001. (**F**) Summary of firing inhibition shown in *pink*, the ability of DAMGO (1 μM) to inhibit ChI firing activity was reduced after pre-desensitization by ME (1 μM) but not deltorphin (1 μM) as compared to the inhibition without pre-desensitization (p < 0.0001 and p = 0.9343, respectively), and in *blue*, the inhibition of firing by deltorphin (1 μM) was attenuated after pre-desensitization by ME (p < 0.0001) but was not different after pre-desensitization by DAMGO (p = 0.6687) as compared to the inhibition without pre-desensitization.

The online version of this article includes the following figure supplement(s) for figure 5:

*Figure 5 continued on next page*

*Figure 5 continued*

**Source data 1.** Desensitization (%) by agonists.

**Source data 2.** Time course analysis during 5 -min application of ME, ME + CTAP and ME + SDM25 N.

**Source data 3.** Inhibition (%) by DAMGO and deltorphin following pre-desensitization as compared to control (no pre-desensitization).

These results suggest that when ME interacts with both MORs and DORs at the same time, MOR desensitization is augmented. To test this assumption, we measured the ability of DAMGO (1 µM) to inhibit the firing rate after ME-induced desensitization (*Figure 5C* bottom trace). The results show a substantial reduction in the inhibition as compared to the control result of DAMGO alone (36.4% ± 6.0% inhibition after ME, n = 7 *vs.* 98.0% ± 1.2% of DAMGO alone, n = 7, one-way ANOVA, p < 0.0001, *Figure 5F* and *Table 1*). In addition, the ability of deltorphin (1 µM) to inhibit the firing rate after ME desensitization was also reduced (18.3% ± 3.4%, n = 5 inhibition after ME *vs.* 96.6 ± 2.7, n = 10 of deltorphin alone, one-way ANOVA, p < 0.0001, *Figure 5F* and *Table 1*). Thus, ME caused desensitization at both MORs and DORs, and desensitization of MORs could be accelerated by DOR desensitization.

## Receptor internalization

One possible explanation for the increase of MOR desensitization when MOR and DOR are simultaneously activated could be co-internalization of both induced by ME. To test for this, we used live cell imaging to examine agonist-induced internalization of endogenous MORs and DORs labeled with NAI-A594 in ChAT-GFP mice. Labeled ChI neurons were first imaged once before agonist application (time = 0 min, *Figure 6A* - top panel). A saturating concentration of each agonist was applied for 20 min followed by a 10 -minute washout before a second image was taken. The control group was treated with only buffer for the same time and images captured at 0 min and 30 min. Fluorescence in the cytoplasm and at the plasma membrane before (F0) and after agonist (F) was measured and used to determine the amount of endocytosis. *Figure 6A* showed the images of cells before (top panel) and after treatment with different agonists and buffer control (bottom panel). The measurement of fluorescence in the cytoplasm and at the plasma membrane was determined by making an outline of the cells based on the GFP signal (*Figure 6B*).

ME (10 µM) caused a visible redistribution of the labeled receptors from plasma membrane into the cytoplasm. Fluorescent puncta were found throughout the cell body of the neuron (*Figure 6A–a*). The F/F0 in the cytoplasm from the ME-treated neurons was significantly higher than F/F0 of cells perfused with buffer alone (1.49 ± 0.05 for ME, n = 9 and 1.13 ± 0.03 for control, n = 8, one-way ANOVA, Dunnett's test, p < 0.0001, *Figure 6C*, top). Deltorphin (1 µM) was used to induce endocytosis of DORs (*Figure 6A–b*). The change of fluorescence in the cytoplasm from deltorphin-treated cells was also greater than in the buffer control group (F/F0 = 1.38 ± 0.04 for deltorphin, n = 7 *vs* 1.13 ± 0.03 for control, n = 8, one-way ANOVA, Dunnett's test, p = 0.0002, *Figure 6C*, top: *Figure 6—source data 1*). Surprisingly, the MOR selective agonist DAMGO (10 µM) caused little or no redistribution of receptors (*Figure 6A-c*), and the F/F0 of the cytoplasm was the same as that of the buffer control group (1.13 ± 0.03, n = 7 DAMGO *vs.* 1.13 ± 0.03 control, n = 8, one-way ANOVA, Dunnett's test, p > 0.9999, *Figure 6C*, top).

Next, the fluorescence at the plasma membrane was determined after each treatment. The fluorescence at the plasma membrane is expected to decrease if labeled receptors are internalized. In control experiments the F/F0 on the plasma membrane was 0.96 ± 0.01 after 30 min (n = 8). A value below one suggests that a small portion of NAI-A594 may bind reversibly and be removed during washing or some small agonist-independent internalization may occur (*Figure 6C*, bottom: *Figure 6—source data 1*). Following the application of ME or deltorphin, the fluorescence at the plasma membrane significantly decreased (F/F0 = 0.85 ± 0.01 for ME, n = 9 and 0.88 ± 0.01 for deltorphin, n = 7; both with p < 0.0001 as compared to control buffer of 0.96 ± 0.01, n = 8, one-way ANOVA, Dunnett's test, *Figure 6C*, bottom). In contrast, DAMGO caused only a slight decrease of the fluorescence ratio at the plasma membrane that was the same as that of the control group (F/F0 = 0.96 ± 0.01, n = 7, one-way ANOVA, Dunnett's test, p = 0.9962, *Figure 6C*, bottom). The results indicate that endogenous DORs are readily internalized whereas MORs remain mostly on the plasma membrane under these experimental conditions. Importantly, ME-induced receptor internalization was comparable to DOR

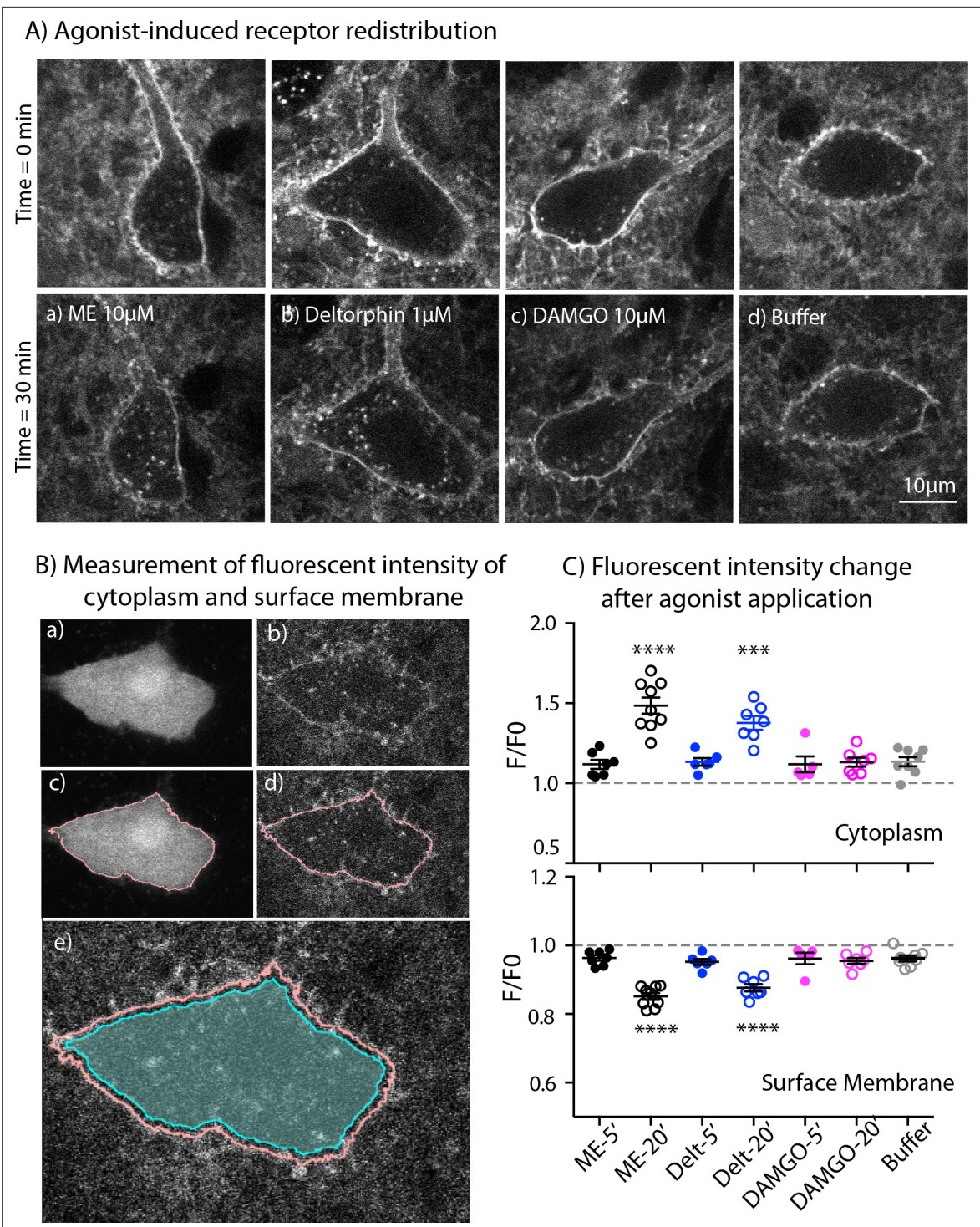

**Figure 6.** Endocytosis of MOR and DOR. (**A**) Images show receptor redistribution induced by different agonists. Upper panel shows images taken before agonist application. Each slice was incubated in NAI-A594 100 nM for 1 hr and washed with Kreb's buffer with continuous bath-perfusion for 10–15 min before acquiring the image. Lower panels show images taken after agonist application by bath-perfusion for 20 min and following 10 min of buffer. The control experiment was treated with buffer alone for 30 min. (**B**) Diagram images show measurements of fluorescent intensities along the plasma membrane and inside the cytoplasm using ImageJ. Raw images of one optical slice of a neuron taken simultaneously are shown for GFP (**a**) and Alexa594 (**b**). The GFP signal is used to define an outline of plasma membrane (**c**). This outline is copied to the image of Alexa594 signals (**d**). The outline of cytoplasm is then dilated eight pixels (~0.6 μm) and the inner line is used to define the area of cytoplasm where internalization is taking place (cyan shade area in (**e**)). Mean fluorescent intensity along the outline of plasma membrane is measured as F-memb (pink line). Mean fluorescent

*Figure 6 continued on next page*

*Figure 6 continued*

intensity in the area of cytoplasm is measured as F-cyt (cyan shade area). The ratio values of F-memb/(F-memb+ F cyt) and F-cyt/(F-memb+ F cyt) are then calculated for F/F0. F is the ratio value after agonist application and F0 is the value before agonist application. (**C**) Summary of fluorescent intensity changes in cytoplasm (top) and plasma membrane (bottom) of neurons treated with agonists. Data shown as mean ± SEM compared to buffer control group, one-way ANOVA, Dunnett's test $p < 0.0001$ (****) and $p = 0.0002$ (***) as compared to control group. ME = [Met$^5$] enkephalin, Delt = deltorphin.

The online version of this article includes the following figure supplement(s) for figure 6:

**Source data 1.** Quantification of receptor internalization at 5 and 20 min.

internalization caused by deltorphin suggesting that ME did not increase MOR endocytosis or cause co-internalization of MOR-DOR.

One caveat for this interpretation is the different time of agonist application between receptor internalization and desensitization. During this period, receptors could undergo endocytosis and recycling that differed from what was happening during the desensitization. Thus, receptor internalization caused by DAMGO (10 μM), deltorphin (1 μM) or ME (10 μM) was measured immediately after 5 -min application of the agonists. The values of F/F0 in the cytoplasm from DAMGO, deltorphin and ME were not different from that of control treated with buffer (1.13 ± 0.03, n = 8 for control; 1.12 ± 0.06, n = 5 for DAMGO, p = 0.9996; 1.13 ± 0.06, n = 6 for deltorphin, p > 0.9999; 1.12 ± 0.05, n = 7 for ME, p = 0.9995, one-way ANOVA, Dunnett's multiple comparisons test as compared to control, *Figure 6C*, top). Similarly, there was no decrease of F/F0 values on the plasma membrane that was different from the buffer control (0.96 ± 0.01, n = 8 for control; 0.96 ± 0.02, n = 5 for DAMGO, p > 0.9999; 0.95 ± 0.01, n = 6 for deltorphin, p = 0.9819; 0.96 ± 0.01, n = 7 for ME, p > 0.9999, one-way ANOVA, Dunnett's test, *Figure 6C*, bottom). The results demonstrate time-dependent processes for deltorphin- and ME-induced receptor internalization and suggest that desensitization occurs before receptor internalization could be detected.

## Discussion

Co-localization of MORs and DORs has been postulated to exist in vivo based on genetically modified receptors in knock-in mice and the use of antibodies (*Gendron et al., 2015*). In the present work, we used receptor imaging and electrophysiological recordings to examine MOR/DOR action and interactions in brain slices from ChAT-GFP and wild-type mice. The results demonstrate that while endogenous MORs and DORs are co-expressed on cholinergic interneurons (ChIs), they function independently and do not appear to require stable dimer formation for their activity.

Interactions between two GPCRs in vivo could result from physical association of the receptors (aka dimerization) or functional consequence of many possible down-stream events (*Lambert and Javitch, 2014*). The discovery that MORs and DORs are present and functional in a single ChI neuron in brain slices offers a relevant model system to study these important issues. The results from this study indicate no major population of stable MOR-DOR heterodimers forms, or if physical association between the receptors does occur, it does not affect the ability of individual receptors to inhibit spontaneous activity of ChIs. In addition, each receptor undergoes distinct regulation processes. DORs desensitize more completely at a faster rate than MORs. Similar results have been reported in studies using cultured neurons from mice with genetic deletions (*Walwyn et al., 2009*), neuroblastoma SH-SY5Y cells expressing endogenous human MORs and DORs (*Prather et al., 1994*) and *Xenopus* oocytes exogenously expressing equal amounts of MOR and DOR (*Lowe et al., 2002*).

When both receptors are concurrently activated with [Met$^5$]-enkephalin (ME), MOR exhibits an increase in desensitization, which differs from the slow induction of desensitization at only MORs. The actions of ME on co-expressed MOR-DOR are crucial in understanding functional interactions between these receptors and resultant effects on ChI activity in vivo. In conditions where ME independently interacts with each receptor (i.e., in the presence of selective antagonists for MOR and DOR), ME produces similar results to the selective agonists thus underscoring the ability of MORs and DORs to act independently in native environment. The increase in MOR desensitization induced by ME alone where MORs and DORs are simultaneously activated is most likely a heterologous action dependent on downstream processes. It is well known that G protein receptor kinase (GRK) phosphorylation and arrestin-binding are common mechanisms for homologous desensitization of MORs and DORs (*Williams et al., 2013*; *Gendron et al., 2016*). Given that ME recruits arrestin to DORs more

efficiently than MORs (*Gomes et al., 2020*), simultaneous co-activation at MOR and DOR by ME may facilitate the interaction between arrestin and MOR. Supporting this possibility, *Lowe et al., 2002* found that in oocyte model, arrestin was the limiting factor for MOR desensitization and increased arrestin expression enhanced the desensitization rate.

One potential issue is that there could be an agonist bias between ME and DAMGO at inducing MOR desensitization. At MOR, ME does not exhibit biased signaling toward G protein or β-arrestin as compared to DAMGO; however, ME does have a biased activity toward β-arrestin at DOR compared to deltorphin (*Gomes et al., 2020*). In the situation when both MORs and DORs were co-activated by ME, recruitment of GRKs/arrestins to the plasma membrane could facilitate MOR desensitization but only when MORs are simultaneously present with DORs. This hypothesis could be tested through the inhibition of GRK2/3 activity and/or genetic deletion of β-arrestin.

Co-internalization of MORs and DORs would provide supporting evidence for physical association of heterodimers. Studies in cultured neurons from mouse dorsal root ganglion (*Walwyn et al., 2009*) and neurons in the tissue of myenteric plexus from DOR-GFP mice (*DiCello et al., 2020*) found no co-internalization induced by a selective agonist of MORs or DORs. Similarly, here we found that internalization of MOR and DOR are very different from each other in ChIs. Our data show deltorphin evoked significant DOR redistribution from the plasma membrane into the cytoplasm, but there was no receptor internalization induced by the MOR selective agonist DAMGO. Because receptor internalization by ME was not greater than that induced by deltorphin, the results suggest that the receptors do not traffic together, even with simultaneous activation. This finding does not eliminate the possibility of transient formation of dimers on the plasma membrane, for example through a so-called 'kiss-and-run' mechanism (*Gurevich and Gurevich, 2008*). In such a scenario, the change in downstream signaling including GRK/arrestin could still modify desensitization that occurs prior to receptor internalization whether the receptors will form dimers or not. It is also not known if the lifetime of a heterodimer will be long enough to have an action on the activation process. Thus far, only MOR-MOR homodimers have been studied by single molecule analyses, and these studies found no evidence of MOR-MOR homodimers at a basal condition, even at high receptor densities (*Möller et al., 2020*; *Asher et al., 2021*). MOR-MOR-homodimers however could be induced by DAMGO with a lifetime of ~460 milliseconds indicating the rapid decay of dimers (*Möller et al., 2020*).

One possible criticism of the above analysis would be to propose that NAI-A594-labeled receptors are less likely to be internalized. We do not think this is likely to be the case for the following reasons. (1) ME induced the same increase in potassium conductance in A594-labeled and unlabeled locus coeruleus neurons (*Arttamangkul et al., 2019*). (2) Agonist-induced endocytosis of NAI-A594 labeled MORs occurs in HEK293 cells (*Arttamangkul et al., 2019*) and in cultured habenular neurons (personal communication with Dr. Damien Jullié, UCSF). (3) The attached fluorophore is of much smaller molecular size compared to reporter proteins or antibodies used in the study of subcellular localization of receptors (*Choquet et al., 2021*), thus it is less likely that A594-tagged receptors would exhibit defective receptor activation. The low signal-to-noise with the use of live brain slices could also limit the detection of low numbers of events and thus accumulation of the fluorescent signal in the cytoplasm. The trafficking of DORs is mainly targeted to lysosomes (*Tsao and von Zastrow, 2000*; *Scherrer et al., 2006*; *Pradhan et al., 2009*), therefore the fluorescent signal in the cytoplasm would increase with time. It should be noted that although the receptors may be degraded in the lysosomes, the Alexa594 dye will remain fluorescent because of its stability in acidic environment (*King et al., 2020*). MORs are known to recycle to the plasma membrane, especially when the agonist remains bound to the receptors (*Yu et al., 2010*; *Jullié et al., 2020*; *Kunselman et al., 2019*). Because slow wash-out of agonist is common in brain slice preparations, it is possible that recycling process of MORs in cultured cells is detected more easily than in brain slices.

Finally, we note our discovery that striatal cholinergic interneurons co-express MORs and DORs, presenting a new perspective on the role of ChIs in regulating synaptic circuitry in the striatum, particularly after opioid treatment. The present study uncovers distribution of MORs and DORs in ChIs throughout the striatum (see *Figure 3D*) and the co-localization of MORs and DORs have been functionally studied in the area of ventral striatum. It is not known if MORs and DORs on ChIs in the dorsal striatum may act like those in the ventral part. The labeling results suggest that MORs and DORs could also co-express on ChIs in the dorsal striatum. Because DORs are more abundant than MORs in the dorsal ChIs (*Figure 3C*), one would predict that MOR desensitization could be accelerated by

co-activation of DORs relative to the observations in the ventral striatum. More importantly, the relative function of MOR and DOR when found on the same cells may be altered after animals are chronically treated with morphine. A number of studies have reported functional up-regulation of DOR in several brain areas, possibly resulting from an increased receptor expression on the surface membrane (reviewed in *Gendron et al., 2015*). In DOR-GFP mice, chronic morphine, however, does not change surface expression of the GFP-tagged DORs on ChIs in the dorsal striatum, and the expression even decreases in the ventral ChIs (*Leah et al., 2016*).

In summary, the results in this study illustrate co-existence of endogenous MOR and DOR on the surface membrane of ChIs in the striatum. We find that both receptors function independently and are differently regulated when activated by selective agonists. In spite of their autonomous signaling, the interactions induced by downstream processes such as phosphorylation and/or arrestin binding suggest important functional interactions apart from direct receptor association.

# Materials and methods

**Key resources table**

| Reagent type (species) or resource | Designation | Source or reference | Identifiers | Additional information |
|---|---|---|---|---|
| Strain, strain background | R.norvegicus, Sprague Dawley, male and female | Charles River Laboratories | Stock# 001 RRID:RGD_734476 | |
| Strain, strain background | *M. musculus*, C57BL/6 J, male and female | Jackson Laboratories | Stock# 000664 RRID:IMSR_JAX:000664 | |
| Strain, strain background (*M. musculus*) | *M. musculus* C57BL/6 J (ChAT-BAC-eGFP) | Jackson Laboratories | Stock # 007902 RRID:IMSR_JAX:007902 | |
| Strain, strain background | *M. musculus* C57BL/6 J MOR-KO | *Schuller et al., 1999* | *Oprm1*-exon-1 knockout mice | Dr. John E. Pintar, RWJMS |
| Cell line | *Homo sapiens* HEK293, female | ATCC | CRL-1573: RRID:CVCL_0045 | Human embryonic kidney |
| Antibody | Anti-Flag-M1 (mouse monoclonal) | Sigma | Cat# F-3040 RRID:AB_439712 | 1:600 |
| Antibody | Anti-Alexa Fluor 488 (rabbit polyclonal) | Thermo Fisher | Cat# A-11094 RRID:AB_221544 | 1:500 |
| Antibody | Goat-anti mouse IgG Alexa 680 | Thermo Fisher | Cat# A28183 RRID:AB_2536167 | 1:5,000 |
| Antibody | Goat-anti rabbit IgG IRDYE CW800 | LI-COR | Cat# P/N 926–32211 RRID:AB_621843 | 1:10,000 |
| Chemical compound, drug | NAI-A594 and NAI-A488 | *Arttamangkul et al., 2019* | N/A | |
| Chemical compound, drug | [Met$^5$]-enkephalin | GenScript | Cat# RP10886 | |
| Chemical compound, drug | Morphine | NIDA | N/A | |
| Chemical compound, drug | CTAP | Sigma | Cat# C6352 | |
| Chemical compound, drug | SDM25N | Tocris | Cat# 1,410 | |
| Chemical compound, drug | Naloxone HCl | Hello Bio | Cat# HB2451 | |
| Chemical, drug | DAMGO | Hello Bio | Cat# HB2409 | |
| Chemical compound, drug | [D-Ala2]-Deltorphin II | Tocris | Cat# 1,180 | |
| Software, algorithm | ScanImage | Vidrio Technologies, LLC | ScanImage RRID:SCR_014307 | |
| Software, algorithm | Fiji Image J | *Rueden et al., 2017* | Fiji RRID:SCR_002285 | |

## Drugs and chemicals

NAI-A594 and NAI-A488 were synthesized as previously described (*Arttamangkul et al., 2019*). 20 nmole of NAI-A594 and NAI-A488 were aliquoted and kept in dried-pellet form at –20 °C until used. A stock solution of 100 µM in 2% DMSO-water was made and used within one week. The following reagents were purchased from companies as indicated in parentheses: DAMGO, naloxone and (+)MK-801 (Hello Bio, Princeton, NJ), [DAla$^2$]-Deltorphin II and SDM25H HCl (Tocris, Minneapolis, MN), [Met$^5$]enkephalin (Genscript, Piscataway, NJ), CTAP (Sigma, St. Louis, MO). Morphine alkaloid was obtained from National Institute on Drug Abuse, Neuroscience Center (Bethesda, MD) and was dissolved in a few drops of 0.1 M HCl and an adjusted volume of water to make a stock solution of 10 mM. All salts to make artificial cerebrospinal fluid (ACSF) used in electrophysiological experiments were purchased from Sigma. Drugs were diluted to the tested concentrations in (ACSF) and applied by bath-perfusion.

## Animals

All animals were handled in accordance with the National Institutes of Health guidelines and with approval from the Institutional Animal Care and Use Committee of OHSU. Mice (21–28 days), both male and female, on a C57BL/6 J background were used for all genotypes. ChAT(BAC)-eGFP transgenic mice were purchased from the Jackson Laboratory (Bar Harbor, ME) and the homozygous pairs were bred. Wild-type mice were raised from breeders obtained from the Jackson Laboratory. MOR-KO (*Oprm1 gene*) mice were gifted from Dr. John Pintar (*Schuller et al., 1999*). Juvenile Sprague-Dawley rats (21–28 days), both male and female, were raised from two breeding pairs that were purchased from Charles River Laboratories (Wilmington, MA).

## Chemical labeling of endogenous MOR and DOR in brain slices

Rats and mice were anesthetized with isoflurane and brains were removed and placed in warm (30 °C) oxygenated ACSF plus MK-801 (0.03 mM). Brain slices were prepared (280 µm thickness) using a vibratome (Leica, Nussloch, Germany). The slices containing striatum between bregma 0.38–1.4 mm of mouse brain and 0.6–2 mm of rat brain were collected and allowed to recover in oxygenated warm ACSF (34 °C) containing 10 µM MK-801 for 30 min. Slices were then hemisected. One half was incubated in oxygenated ACSF containing 100 nM NAI-A594 for 1 hr at room temperature. In a separate vial, the other half of brain slice was also incubated for 1 hr at room temperature in the solution of 100 nM NAI-A594 plus CTAP (1 µM) or NAI-A594 plus SDM25N (0.5 µM) to selectively block the labeling of MOR or DOR, respectively. The labeled slice was then transferred to an upright microscope (Olympus BX51W1, Center Valley, PA) equipped with a custom-built two-photon apparatus and a 60 x water immersion lens (Olympus LUMFI, NA 1.1). Both Alexa594 dye and GFP were excited simultaneously with a high-frequency laser beam at 810 nm (Chameleon, Coherent, Inc, Santa Clara, CA). Fluorescence of GFP and Alexa594 were acquired and collected in two separate channels using ScanImage software (*Pologruto et al., 2003*). Each image acquisition contained 10 optical slices at 0.5 µm thickness. Each optical slice was averaged from five consecutive scans of 512 × 512 pixels. Brain slices were submerged in continuous flow of ACSF at a rate of 1.5–2 ml/min and drugs were applied via superfusion. All experiments were done at 34 °C. Macroscopic images of labeled slices were captured with a Macro Zoom Olympus MVX10 microscope and a MV PLAPO2xC, NA 0.5 lens (Olympus). Alexa594 dye was excited with yellow LED (567 nm). The slices were kept under ACSF in a petri dish during data acquisitions. Data were acquired using Q Capture Pro software (Q Imaging, British Columbia, Canada).

## Electrophysiology

Brain slices were prepared as described for the labeling study. A loose cell-attached extracellular recording was done with an Axopatch-1C amplifier in current mode. Recording pipettes (1.7–2.1 MΩ, TW150F-3, World Precision Instruments, Saratosa, FL) were filled with 175 mM NaCl buffered to pH7.4 with 5 mM HEPES. The on-cell pipet resistance varied in the range of 5–15 MΩ. Immediately the cell attaching was formed, spontaneous activity of the neuron was monitored. Only cells that showed steady regular firing at baseline for 3–5 min were used in the experiments. Data were collected at sampling rate of 10 KHz, and episode width of 50 s with Axograph X (1.5.4). Drugs were applied by

bath perfusion. Bestatin (10 µM, Sigma) and thiorphan (1 µM, Sigma) were included in all peptide solutions to prevent enzymatic degradation.

## Chemical labeling of FDOR and FMOR in HEK293 cells

Human embryonic kidney 293 (HEK293) cells were originally obtained from ATCC (Cat. No. CRL_1573) and maintained in house. Its identity has been authenticated and there is no mycoplasma contamination. HEK293 cells expressing FlagDOR and FlagMOR were generated by transfection with FlagMOR and FlagDOR plasmids obtained from Dr. Mark von Zastrow (UCSF). Stable expressing cells were selected, grown and maintained in Dulbecco's minimal essential media (DMEM, Gibco, Grand Island, NY) containing 10 % fetal bovine serum (FBS), Geneticin sulfate (0.5 mg/ml, ThermoFisher, Waltham, MA) and antibiotic-antimycotic (1 x, ThermoFisher). Twenty-four hr before labeling, cells were seeded at $3.0 \times 10^5$ cells/well in a 12-well plate. The next day, cells were labeled with 30 or 100 nM NAI-A488 for 30 min at 37 °C. Control cells were labeled in the presence of excess 10 µM naloxone to confirm the specificity of labeling. After the labeling period, cells were washed two times in PBS + 1 µM naloxone, followed by incubation with 10 µM naloxone in PBS containing calcium and magnesium at 37 °C for another 10 min. Cells were then placed on ice and washed two times in ice-cold PBS plus 1 µM naloxone. The wells were scraped, and cell pellets collected by centrifugation at 4000 rpm for 4 min at 4 °C in an Eppendorf microcentrifuge. Whole cell extracts were produced by resuspending cell pellets in lysis buffer (50 mM Tris pH 7.4, 150 mM NaCl, 1.0 % nonidet P-40, 0.5% sodium deoxycholate, 0.5 mM PMSF, 5 µg/ml leupeptin, 1 X Halt protease inhibitor cocktail (Thermo Scientific Cat. No. 78437)) and nutating for 30 min at 4 °C, followed by centrifugation at 14,000 rpm for 20 min at 4 °C. The supernatant was collected for further analysis. Protein concentrations were determined using Bio-Rad DC Protein Assay Kit II (cat. no. 5000112) according to the manufacturer's recommendations. The NAI-A488-labeled cell extracts were then subjected to western blotting.

## Western blotting

Cell extracts prepared from NAI-A488-labeled cells were analyzed by SDS-PAGE followed by western immunoblotting. Ten µg total protein of each sample was denatured in SDS containing sample buffer (2 % SDS, 10 % glycerol, 62.5 mM Tris pH 6.8, 0.001 % bromophenol blue) in the presence of 0.1 M DTT, at 37 °C for 10 min. The samples were then loaded onto 8 % SDS-polyacrylamide denaturing gels and run until the dye front just ran off the bottom. The proteins were then electro-transferred to PVDF membranes (Immobilon-FL, cat. no. IPFL00010), blocked in Odyssey Blocking Buffer (LI-COR, cat. no. 927–40000) and probed with two primary antibodies, M1 anti-FLAG MAb (Sigma, cat. no. F3040, used at a 1:600 dilution) and rabbit anti Alexa Fluor 488 (Invitrogen, cat. no. A11094, 1:500), followed by washing and incubation with two secondary antibodies, goat-anti mouse IgG Alexa-680 (Thermo Fisher, A28183, 1:5,000 dilution) and goat-anti rabbit IgG IRDye CW 800 (LI-COR cat no. P/N 926–32211, 1:10,000 dilution). Antibodies were diluted in a buffer consisting of 1/10 diluted Odyssey Blocking Buffer, TBS (20 mM Tris pH 7.4, 100 mM NaCl)/1 mM $CaCl_2$ and 0.2 % Tween-20. Blots were washed in TBS/1 mM $CaCl_2$/0.3 % Tween-20, and final washes were in TBS/1 mM $CaCl_2$. The western blot membranes were scanned using a Sapphire Biomolecular Imager (Azure Biosystems) at 784 nm and 658 nm excitation and images were acquired using Sapphire Capture software (2017).

## On-cell western analysis

96-well, black walled, optical bottom plates (Thermo Fisher Scientific, cat. no. 165305) were coated with 0.1 mg/ml poly-D-lysine followed by seeding, in triplicate for each condition, $3.0 \times 10^4$ DOR or MOR cells per well. 24 hr later, using the same method as above, the cells were labeled with varying concentrations of NAI-A488 (0, 1, 3, 10, 30, 100 nM) and duplicate wells were labeled with each NAI-A488 concentration in the presence of 10 µM naloxone, in order to determine non-specific binding. After the final PBS wash, cells were fixed in 3.7 % formaldehyde, 20 min, at room temperature, washed in PBS three times and then blocked for 1.5 hr, at room temperature, with rocking, in Odyssey Blocking Buffer. The fixed and blocked cells were then incubated overnight at 4 °C, on an orbital shaker with primary antibodies (M1, 1:600 and rabbit anti-Alexa Fluor 488, 1:500) diluted in 1:10 Odyssey Blocking Buffer in TBS/1 mM CaCl2. The next morning, the wells were washed five times, 5 min each in TBS containing 1 mM $CaCl_2$ and 0.05 % Tween-20 (OCW wash buffer), followed by incubation in secondary antibodies (Goat-anti mouse IgG-Alexa-680, 1:2500; goat anti-rabbit-IRDye

CW 800, 1:5000 diluted in OCW wash buffer) for 1 h, at room temperature while rocking. The wells were then washed five times for 5 min in wash buffer, followed by three washes for 5 min each in TBS plus 1 mM CaCl$_2$. The wells were air-dried before imaging on a Sapphire Biomolecular Imager using 784 nm and 658 nm excitation. In this experiment, the signal from 784 nm excitation represents opioid receptors modified by NAI-A488, while the 658 nm signal indicates opioid receptor present in each condition. Background values were obtained from cells that were labeled with 100 nM NAI-A488, but that did not receive any primary antibody. Fluorescence intensities were quantitated using AzureSpot Analysis Software (version 2.0.062). All values were background subtracted and the 784 nm signal for each condition was normalized to the 658 nm signal in the same condition. The ratios (784/658) were also calculated for cells labeled with NAI-A488 in the presence of 10 µM naloxone. Prism 6 (GraphPad software, SanDiego, CA) was used to construct concentration dependent curves of non-specific and total labeling and calculate Kd's.

## Quantitative image analysis of membrane fluorescent intensity

We wrote an ImageJ macro to automate the fluorescent intensity analysis of two-photon microscopy data using Fiji software (*Rueden et al., 2017*). The data were collected from two fluorescence emission channels, GFP and Alexa 594. The macro inputs a data stack, first deinterleaving the channels and Gaussian blurring (sigma = 1) to smooth the GFP channel's pixels (*Figure 3B*). Moments-preserving thresholding is then applied to generate a mask on each slice of the GFP signal. We assigned this GFP mask's edge corresponding to the shape of the neuron (*Figure 3B*). The assignment was verified qualitatively, and the result was well-matched with the signals of NAI-A594 staining on the membrane. Some errors however may occur particularly along dendritic branching planes resulting in truncated masking (down skewing the membrane mean intensity measurement), and the data in this area would be avoided (see *Figure 3B*). The mask from each GFP optical slice was then overlaid onto the same optical slice of corresponding Alexa594 channel. The procedure was scanned through every optical section to ensure that the largest area corresponding to the shape of the neuron was included in the analysis. The membrane fluorescent intensity was measured by converting the GFP mask area to a line with a random break introduced along its length. The line traced along the membrane-mask edge and thereby the neuron's membrane (*Figure 3B*). The mean pixel intensity was collected along the line's length. Each data stack is analyzed over ten consecutive slices. The mean fluorescence of each optical slice was pooled and computed for the mean membrane fluorescence intensity of a neuron (F-memb). The macro was run without intervention by a researcher who was blinded to the data collection.

## Quantitative image analysis of receptor endocytosis

With the same masking procedure, the analysis comprised two separate operations on the selected mask. The first operation was to determine the mean membrane fluorescent intensity (F-memb) as described earlier. The second operation used an 8-pixel dilation (corresponding to ~0.6 µm) along every point in the membrane mask's definition to define an area of the neuron's cytoplasmic portion (*Figure 6B*). The mean cytoplasmic fluorescent intensity of each optical slice was measured from this area, and all slices' means were pooled and averaged for the final output of mean cytoplasmic fluorescence (F-cyt) of a neuron.

Receptor endocytosis was determined by two methods. One method calculated the change of cytoplasmic fluorescent intensity from the ratio of F/F0, where F = F-cyt/(F-cyt+ F memb) after drug treatment and F0 = F-cyt/(F-cyt+ F memb) before drug treatment. The other method used F/F0 calculated from membrane fluorescent intensity and F = F-memb/(F-cyt+ F memb) after drug and F0 = F-memb/(F-cyt+ F memb) before drug.

## Statistical analysis

Data are presented as mean ± standard error of the mean (SEM) except the data of fluorescent membrane surface intensity, which is shown as mean ± standard deviation (SD). The number of sample size is indicated in the figure legends. Statistical analysis and graphs were made with Prism 6 (GraphPad Software, SanDiego, CA). One-way ANOVA with Sidak's multiple comparison test was used to compare treatment groups and $p < 0.05$ was considered significant. A non-linear saturation of one-site binding curve was constructed from total and non-specific labeling curve and Kd was calculated from curve fitting values using Prism 6.

## Acknowledgements

The authors thank Dr. John T Williams for his constructive advice on the project and manuscript and Dr. Marina E Wolf for her critical comments on the manuscript. We also thank Drs. Aaron Nilsen and Victoria S Halls at the Medicinal Chemistry Core, OHSU, for the synthesis of fluorescent NAI compounds and Dr. Kenner Rice at Drug Design and Synthesis Section, NIDA and NIAAA for providing naltrexamine. The work is funded by grant DA048136 from the National Institutes of Health.

## Additional information

### Funding

| Funder | Grant reference number | Author |
| --- | --- | --- |
| National Institute on Drug Abuse | DA048136 | Seksiri Arttamangkul David Farrens |

The funders had no role in study design, data collection and interpretation, or the decision to submit the work for publication.

### Author contributions

Seksiri Arttamangkul, Conceptualization, Data curation, Formal analysis, Funding acquisition, Investigation, Methodology, Project administration, Resources, Supervision, Visualization, Writing - original draft, Writing - review and editing; Emily J Platt, Data curation, Formal analysis, Investigation, Methodology, Validation, Writing - review and editing; James Carroll, Formal analysis, Software, Writing - review and editing; David Farrens, Investigation, Methodology, Supervision, Validation, Writing - review and editing

### Author ORCIDs

Seksiri Arttamangkul http://orcid.org/0000-0002-8815-5124
Emily J Platt http://orcid.org/0000-0001-8128-4751
James Carroll http://orcid.org/0000-0002-9264-4502

### Ethics

All animal uses were conducted in accordance with the National Institutes of Health guidelines and with approval from the Institutional Animal Care and Use Committee (IACUC) protocol #IP00000160 of the Oregon Health & Science University. Rats and mice were anesthetized with isoflurane before euthanized with minimal suffering.

### Decision letter and Author response

Decision letter https://doi.org/10.7554/69740.sa1
Author response https://doi.org/10.7554/69740.sa2

## Additional files

### Supplementary files

• Transparent reporting form

### Data availability

All data generated or analyzed during this study are included in the manuscript.

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
