## [Decision Letter]

**Acceptance summary:**

This manuscript addresses an important question in the opioid field – whether the mu opioid receptor and the δ opioid receptor are likely to function independently, or whether their signaling is coupled, e.g. if they form heterodimers. Using live imaging with a tag that labels both receptors and electrophysiological recordings in cholinergic striatal interneurons in brain slices, the authors conclude that the receptors most likely act independently. The work is carefully done, and while the possibility remains that there may be interactions between the two receptors, either by rapid on/off dimerization or downstream signaling, this work adds to our understanding of the system and provides important evidence in favor of independent receptor activation.

**Decision letter after peer review:**

Thank you for submitting your article "Functional independence of endogenous μ- and δ-opioid receptors co-expressed in cholinergic interneurons" for consideration by *eLife*. Your article has been reviewed by 3 peer reviewers, one of whom is a member of our Board of Reviewing Editors, and the evaluation has been overseen by Lu Chen as the Senior Editor. The following individual involved in review of your submission has agreed to reveal their identity: Brady K Atwood (Reviewer #3).

Essential revisions:

1. The manuscript lacks detailed statistical analyses, mostly relying on descriptions of the data as interpreted by the authors. In most places there are no indications of which statistical analyses were utilized and what the outcomes of those analyses were. In the few places where analyses were indicated, it is not clear that the appropriate tests were used (e.g. using an unpaired t-test when an ANOVA, and possibly a repeated measures ANOVA should have been used).

2. The Met-enkephalin data are puzzling and raise several concerns.

a) 5D middle trace looks like ME-induced desensitization could just be slower for MOR, please show a longer time course than 5 min in 5E.

b) 5C: the order of treatments is highly inconsistent (agonist-agonist, agonist-antagonist-agonist, etc). Order seems to be important as DAMGO immediately after Delt doesn't desensitize (top panel in 5C), but does when applied first (middle panel of 5C), but is less efficacious following a washout of MetEnk (lower panel of 5C). In order to make this interpretable, please use a consistent approach to treatments: how long they are on, what treatments are used between DAMGO/Met-Enk and Delt/Met-Enk, etc.

c) It remains unclear why Met-enk pretreatment results in MOR desensitization. The authors hypothesize that the desensitization that occurs in MORs after Met-enk administration could be due to the coincidental recruitment of GRK/arrestin by DORs. However, the data in Figure 6 suggests that Met-enk does not result in greater internalization, which would argue against this hypothesis. Please provide more data that establish this interpretation.

d) Address why Met-Enk is different than DAMGO. Is it biased signaling? (try a highly arrestin-biased agonist or in an arrestin KO tissue)? Met-Enk was used to co-activate the two receptors, why not try a co-activation with DAMGO and Delt as a parallel to the Met-Enk. Same results or different?

*Reviewer #1 (Recommendations for the authors):*

The manuscript by Arttamangkul et al., tests how the responses to a selective agonist to one receptor affects subsequent responses to an agonist of the other, using receptor internalization and electrophysiological responses to gauge the likelihood that MOR and DOR receptors act independently vs. as a unit. The work is carefully done, and the authors conclude that the two receptors act independently in these neurons. However, the work cannot exclude that some of the receptors also act together.

Suggestions for revision:

1. While the agonists used in the present study likely target both pathways in the MOR and DOR, it remains possible that in a heterodimer, signaling might instead be biased. There may only be a subset of heterodimers or MOR/DOR interactions of some other sort, but it is difficult to rule this out with the approaches here.

2. Why in Figure 5C/F was CTAP added prior to applying deltorphin? If the point of the experiment is to detect whether the two agonists have additive effects, adding a competitive antagonist seems counterintuitive.

3. Met-enkephalin produced desensitization of MORs that could be accelerated by desensitization of DORs; how do the authors rule out that this finding results from interaction/dimerization between the two?

4. The authors indicate that the different rates of desensitization with MOR vs DOR agonists must mean that the two are not interacting. It seems to me that distinct conformations of a single heterodimer could yield the same result. Please clarify.

*Reviewer #2 (Recommendations for the authors):*

Justified Conclusions:

1) MORs and DORs are both expressed on ChIs.

2) MORs and DORs inhibit ChI firing.

3) DORs desensitize more robustly than MORs with saturating agonist concentrations.

4) MOR and DOR inhibition of ChIs is mediated through parallel mechanisms.

5) Met-enk facilitates desensitization of MORs.

6) DORs primarily desensitize via internalization. MORs do not.

Unjustified Conclusions:

1) It remains unknown how MORs and DORs mediate endogenous Met-enk signaling in dorsal striatum.

2) The authors hypothesize that the desensitization that occurs in MORs after Met-enk administration could be due to the coincidental recruitment of GRK/arrestin by DORs. However, the data in Figure 6 suggests that Met-enk does not result in greater internalization, which would argue against this hypothesis. The authors should provide more data that establish this interpretation.

Suggestions:

1) I think the authors meant to call out Figure 5A in line 247.

2) The GRK/arrestin hypothesis is intriguing. Although not necessary for this study, the proposed experiment in the discussion of using arrestin KO mice would be of great value. I suggest removing that data until it can be included with more rigorous experiments, or do the better experiment to definitively prove this conclusion.

3) The authors use the term dimerization as a very accepted term, when in fact the very nature of their study shows that at best there is barely an interaction at the level of signaling. The authors need to consider the term more deeply (think pentameric ion channels as the bar for direct protein-protein dimers), and revise the language, intro and discussion throughout to better clarify what they mean by GPCR dimers? Do they mean like GABA-B, or do they mean like ion channels…or do they mean like many people think Class A receptors that kiss one another quickly, and then move on. And/or that the receptors are packaged together for anterograde transport or retrograde trafficking and that's really all they come close to one another for. There are many assumptions at the outset that need more careful language so that the field can read this paper with the correct frame of mind and healthy skepticism, which the authors are trying to bring forth.

Questions:

1) Why does DOR antagonism revert ChI firing back to baseline more quickly than CTAP after DAMGO? This seems counterintuitive since the DORs appear to use an internalization mechanism for desensitization, in which case receptor antagonism should be less effective.

2) Considering the baseline differences in likelihood of desensitization, does this suggests that DORs are more effective GRK recruiters than MORs? Could this further suggests that DORs preferentially act via Gbeta/gama, whereas MORs act through Galpha mechanisms?

---

## [Author Response]

Essential revisions:1. The manuscript lacks detailed statistical analyses, mostly relying on descriptions of the data as interpreted by the authors. In most places there are no indications of which statistical analyses were utilized and what the outcomes of those analyses were. In the few places where analyses were indicated, it is not clear that the appropriate tests were used (e.g. using an unpaired t-test when an ANOVA, and possibly a repeated measures ANOVA should have been used).

Statistical analyses are now included in all the results.

2. The Met-enkephalin data are puzzling and raise several concerns.

The results with Met-enkephalin (ME) are very interesting. All the concerns are addressed below.

a) 5D middle trace looks like ME-induced desensitization could just be slower for MOR, please show a longer time course than 5 min in 5E.

This is a valid point. Our data show that MOR desensitization caused by ME + SDM25N was slower than desensitization of both receptors induced by ME alone or DOR desensitization induced by ME + CTAP during 5-minute desensitization (two-way ANOVA for time treatment, p = 0.0040 or p<0.000, respectively). The analysis results have been included in the result section.

Although in principle we agree that a time course longer than 5 minutes could show values of MOR desensitization closer to that of DOR desensitization, we have not carried out the suggested experiment for two reasons. First, we do not think that the results would change the conclusion of our study, as it would not change the fact that we observe different regulation processes of MORs and DORs co-expressed in a single neuron. Second, measuring spontaneous firing in real-time is technically challenging experiment to carry out, especially when we do not know how long we would have to measure (since we do not know when MOR desensitization will reach the same extent as DOR). Hence, we do not include these experiments but instead included the results of MOR and DOR desensitization selectively induced by ME that were analyzed by two-way ANOVA (see above and line 275-277 in the revised manuscript).

b) 5C: the order of treatments is highly inconsistent (agonist-agonist, agonist-antagonist-agonist, etc). Order seems to be important as DAMGO immediately after Delt doesn't desensitize (top panel in 5C), but does when applied first (middle panel of 5C), but is less efficacious following a washout of MetEnk (lower panel of 5C). In order to make this interpretable, please use a consistent approach to treatments: how long they are on, what treatments are used between DAMGO/Met-Enk and Delt/Met-Enk, etc.

To address this, we did experiments where there was no antagonist between the application of DAMGO (10 µM, 5 minutes) and deltorphin (1 µM, 2 minutes). The results were not different from those that used CTAP. This new experiment and results are now presented in Figure 5C and 5 F.

In the top trace, there was no DAMGO (1 µM)-induced desensitization because of a short application time (2 min). This application is very different from the middle trace where application of a higher concentration of DAMGO, 10 µM was applied for 5 minutes.

The results on the top and middle trace in Figure 5C and the summary (Figure 5F) indicated that pre-desensitization of MOR by DAMGO or DOR by deltorphin did not reduce the subsequent inhibition of firing caused by activation of -DOR or MOR, respectively.

The bottom trace shows that the ME caused desensitization of the combination of MOR and DOR. The inhibition induced by DAMGO or deltorphin was reduced following washout of ME (summary is included in Figure 5F).

c) It remains unclear why Met-enk pretreatment results in MOR desensitization. The authors hypothesize that the desensitization that occurs in MORs after Met-enk administration could be due to the coincidental recruitment of GRK/arrestin by DORs. However, the data in Figure 6 suggests that Met-enk does not result in greater internalization, which would argue against this hypothesis. Please provide more data that establish this interpretation.

To address this concern, we did experiments where internalization was examined after 5-minute treatment with ME, deltorphin or DAMGO to compare with the desensitization time course. There was negligible internalization by any agonist. The data is included in Figure 6C. These results suggest that receptor desensitization occurred prior to internalization and that GRK/arrestin recruitment results in desensitization without internalization.

d) Address why Met-Enk is different than DAMGO. Is it biased signaling? (try a highly arrestin-biased agonist or in an arrestin KO tissue)?

DAMGO is a selective MOR agonist. ME is a full agonist at MOR and DOR. DAMGO and ME are non-biased agonists at MOR while ME shows small biased activity toward b-arrestin at DOR (Gomes et al., 2020). We reason that the GRK/arrestin pathway may play important roles for this difference between MOR and DOR and that we have planned to address this question in the future work using GRK inhibitors as well as arrestin KO tissues.

Met-Enk was used to co-activate the two receptors, why not try a co-activation with DAMGO and Delt as a parallel to the Met-Enk. Same results or different?

This study focused on ME because it is an endogenous peptide that activates both receptors. Using the same ligand will provide better understanding of biased signaling by different receptors in the same environment. It is also more physiological relevant because the D2-MSNs in striatum produce enkephalin.

Reviewer #1 (Recommendations for the authors):The manuscript by Arttamangkul et al., tests how the responses to a selective agonist to one receptor affects subsequent responses to an agonist of the other, using receptor internalization and electrophysiological responses to gauge the likelihood that MOR and DOR receptors act independently vs. as a unit. The work is carefully done, and the authors conclude that the two receptors act independently in these neurons. However, the work cannot exclude that some of the receptors also act together.Suggestions for revision:1. While the agonists used in the present study likely target both pathways in the MOR and DOR, it remains possible that in a heterodimer, signaling might instead be biased. There may only be a subset of heterodimers or MOR/DOR interactions of some other sort, but it is difficult to rule this out with the approaches here.

This concern has been addressed in the public review.

2. Why in Figure 5C/F was CTAP added prior to applying deltorphin? If the point of the experiment is to detect whether the two agonists have additive effects, adding a competitive antagonist seems counterintuitive.

We used CTAP to reverse the inhibition of firing by DAMGO to baseline before application of deltophin. To address any concerns about this protocol, we now included new results where CTAP was not used, and show that deltorphin was able to inhibit the firing the same as without pre-desensitization with DAMGO.

3. Met-enkephalin produced desensitization of MORs that could be accelerated by desensitization of DORs; how do the authors rule out that this finding results from interaction/dimerization between the two?

We rule out the possible stable dimer formation based on the finding that MORs and DORs are not co-internalized by ME. The transient heterodimer formation is possible but experiments to determine the dynamic interaction between the two receptors at molecular levels such as FRET-FLIM will be necessary. This will require a selective labeling at MOR and DOR with a donor-acceptor pair of fluorophores by NAI reagents, which is beyond the scope of this current work.

4. The authors indicate that the different rates of desensitization with MOR vs DOR agonists must mean that the two are not interacting. It seems to me that distinct conformations of a single heterodimer could yield the same result. Please clarify.

The results from ME induced desensitization suggested that the two receptors did not form a distinct conformation. If both of the MOR and DOR binding sites were available in this distinct conformation, DAMGO or deltorphin would desensitize the receptor at the same rate since a dimer would act as a single unit. If one of the binding sites is not accessible to the agonist, ME + CTAP or ME + SDM25 would show the same desensitization rate because ME would be able to bind either MOR or DOR. Our results do not support the assumption of a single dimerization unit.

Reviewer #2 (Recommendations for the authors):Justified Conclusions:1) MORs and DORs are both expressed on ChIs.2) MORs and DORs inhibit ChI firing.3) DORs desensitize more robustly than MORs with saturating agonist concentrations.4) MOR and DOR inhibition of ChIs is mediated through parallel mechanisms.5) Met-enk facilitates desensitization of MORs.6) DORs primarily desensitize via internalization. MORs do not.Unjustified Conclusions:1) It remains unknown how MORs and DORs mediate endogenous Met-enk signaling in dorsal striatum.

We agree for this comment. This study focused on the functions of MOR and DOR in the ventral striatum since the expression of MORs and DORs as determined by NAI labeling in this area was not different. The data from NAI-A594 labeling showed that there were more DORs than MORs in the dorsal than ventral striatum, thus it is possible that action of MOR and DOR in the in the dorsal striatum could be different from the ventral striatum. This is something that we are interested to investigate in the future.

2) The authors hypothesize that the desensitization that occurs in MORs after Met-enk administration could be due to the coincidental recruitment of GRK/arrestin by DORs. However, the data in Figure 6 suggests that Met-enk does not result in greater internalization, which would argue against this hypothesis. The authors should provide more data that establish this interpretation.

The internalization data were collected after 20 min application of the agonists while the desensitization occurred within 3-5 min. We added the results of agonist-induced internalization for 5 minutes for comparison. There was negligible receptor internalization caused by ME, deltorphin or DAMGO, and that a longer incubation time were needed to detect receptor internalization. The results show that desensitization precedes internalization.

Suggestions:1) I think the authors meant to call out Figure 5A in line 247.

Thank you for spotting this mistake. We corrected it in the revised manuscript.

2) The GRK/arrestin hypothesis is intriguing. Although not necessary for this study, the proposed experiment in the discussion of using arrestin KO mice would be of great value. I suggest removing that data until it can be included with more rigorous experiments, or do the better experiment to definitively prove this conclusion.

There is no data to date using arrestin KO.

3) The authors use the term dimerization as a very accepted term, when in fact the very nature of their study shows that at best there is barely an interaction at the level of signaling. The authors need to consider the term more deeply (think pentameric ion channels as the bar for direct protein-protein dimers), and revise the language, intro and discussion throughout to better clarify what they mean by GPCR dimers? Do they mean like GABA-B, or do they mean like ion channels…or do they mean like many people think Class A receptors that kiss one another quickly, and then move on. And/or that the receptors are packaged together for anterograde transport or retrograde trafficking and that's really all they come close to one another for. There are many assumptions at the outset that need more careful language so that the field can read this paper with the correct frame of mind and healthy skepticism, which the authors are trying to bring forth.

The term of dimerization has been included in the introduction and discussion. Many class A GPCRs, can associate transiently.

Questions:1) Why does DOR antagonism revert ChI firing back to baseline more quickly than CTAP after DAMGO? This seems counterintuitive since the DORs appear to use an internalization mechanism for desensitization, in which case receptor antagonism should be less effective.

DOR antagonism reversed the inhibition of ChI firing more quickly because of the combination of antagonism and desensitization. The inhibition induced by DAMGO, was more sustained resulting from less receptor desensitization such that DAMGO had to dissociate before the antagonist binding. Moreover, our new results show no DOR internalization is detected after 5-minute agonist application when a significant desensitization occurs. These results indicate that DOR internalization is separable from desensitization.

2) Considering the baseline differences in likelihood of desensitization, does this suggests that DORs are more effective GRK recruiters than MORs? Could this further suggests that DORs preferentially act via Gbeta/gama, whereas MORs act through Galpha mechanisms?

Regarding to the first question, we believe it is possible that DORs are more effective GRK recruiters as DORs underwent endocytosis better than MORs. We plan to test the sensitivity of compound 101 (GRK2/3 inhibitor) to inhibit the GRK recruitment by MOR and DOR. For the second question, there is no evidence to date that the inhibition of firing induced by MORs is dependent on Galpha subunit mechanism.